# Untargeted Metabolomics by Ultra-High-Performance Liquid Chromatography Coupled with Electrospray Ionization-Quadrupole-Time of Flight-Mass Spectrometry Analysis Identifies a Specific Metabolomic Profile in Patients with Early Chronic Kidney Disease

**DOI:** 10.3390/biomedicines11041057

**Published:** 2023-03-30

**Authors:** Mihaela-Roxana Glavan, Carmen Socaciu, Andreea Iulia Socaciu, Florica Gadalean, Octavian M. Cretu, Adrian Vlad, Danina M. Muntean, Flaviu Bob, Oana Milas, Anca Suteanu, Dragos Catalin Jianu, Maria Stefan, Lavinia Balint, Silvia Ienciu, Ligia Petrica

**Affiliations:** 1Department of Internal Medicine II—Nephrology, “Victor Babeș” University of Medicine and Pharmacy, 300041 Timișoara, Romania; 2Centre for Molecular Research in Nephrology and Vascular Disease, Faculty of Medicine, “Victor Babeș” University of Medicine and Pharmacy, 300041 Timișoara, Romania; 3Research Center for Applied Biotechnology and Molecular Therapy BIODIATECH, SC Proplanta, 400478 Cluj-Napoca, Romania; 4Department of Occupational Health, University of Medicine and Pharmacy “Iuliu Haţieganu”, 400347 Cluj-Napoca, Romania; 5Department of Surgery—Surgical Semiotics, “Victor Babeş” University of Medicine and Pharmacy, 300041 Timişoara, Romania; 6Department of Internal Medicine II—Diabetes and Metabolic Diseases, “Victor Babeș” University of Medicine and Pharmacy, 300041 Timișoara, Romania; 7Department of Functional Sciences—Pathophysiology, Faculty of Medicine, “Victor Babeș” University of Medicine and Pharmacy, 300041 Timișoara, Romania; 8Center for Translational Research and Systems Medicine, Faculty of Medicine, “Victor Babeș” University of Medicine and Pharmacy, 300041 Timișoara, Romania; 9Deptartment of Neurosciences—Neurology, “Victor Babeș” University of Medicine and Pharmacy, 300041 Timișoara, Romania; 10Centre for Cognitive Research in Neuropsychiatric Pathology, Clinical County Emergency Hospital, Victor Babeș” University of Medicine and Pharmacy, 300723 Timișoara, Romania

**Keywords:** chronic kidney disease, metabolomics, biomarkers, amino acids, uremic toxins, acylcarnitines

## Abstract

Chronic kidney disease (CKD) has emerged as one of the most progressive diseases with increased mortality and morbidity. Metabolomics offers new insights into CKD pathogenesis and the discovery of new biomarkers for the early diagnosis of CKD. The aim of this cross-sectional study was to assess metabolomic profiling of serum and urine samples obtained from CKD patients. Untargeted metabolomics followed by multivariate and univariate analysis of blood and urine samples from 88 patients with CKD, staged by estimated glomerular filtration rate (eGFR), and 20 healthy control subjects was performed using ultra-high-performance liquid chromatography coupled with electrospray ionization-quadrupole-time of flight-mass spectrometry. Serum levels of Oleoyl glycine, alpha-lipoic acid, Propylthiouracil, and L-cysteine correlated directly with eGFR. Negative correlations were observed between serum 5-Hydroxyindoleacetic acid, Phenylalanine, Pyridoxamine, Cysteinyl glycine, Propenoylcarnitine, Uridine, and All-trans retinoic acid levels and eGFR. In urine samples, the majority of molecules were increased in patients with advanced CKD as compared with early CKD patients and controls. Amino acids, antioxidants, uremic toxins, acylcarnitines, and tryptophane metabolites were found in all CKD stages. Their dual variations in serum and urine may explain their impact on both glomerular and tubular structures, even in the early stages of CKD. Patients with CKD display a specific metabolomic profile. Since this paper represents a pilot study, future research is needed to confirm our findings that metabolites can serve as indicators of early CKD.

## 1. Introduction

Chronic kidney disease (CKD) is a progressive disease with increased mortality and high prevalence, affecting 11.7–15.1% of the population in developed countries, and has emerged as one of the most significant progressive diseases of the twenty-first century [1]. A specific clinical and biological pattern and multiple comorbidities characterize CKD. As a result, identification of the underlying pathogenic mechanisms and of possible treatment options has become a challenge for modern medicine. CKD is a condition characterized by a wide range of pathological abnormalities, from kidney damage with normal kidney function to end-stage renal disease (ESRD). Hemodynamic and metabolic factors, inflammation, proteinuria, podocyte loss, proximal tubule dysfunction, dyslipidemia, the renin-angiotensin-aldosterone system, and oxidative stress represent the underlying mechanisms involved in CKD development and progression [2].

The classification of CKD is based on the evaluation of kidney damage markers (albuminuria) and estimated glomerular filtration rate (eGFR). In clinical practice, measurement of serum creatinine concentration is mandatory for eGFR assessment. There are certain drawbacks to this approach, such as the fact that serum creatinine levels vary with muscle mass and age [3]. On the other hand, fever and infection can cause kidney damage and increase albuminuria. Therefore, the resolution of kidney dysfunction does not correspond with the resolution of albuminuria and can increase the risk of unnecessary treatment in patients with underlying kidney pathologies [4]. Therefore, the discovery of new markers for earlier and more accurate diagnosis of CKD and new therapeutic options have become mandatory.

Omics sciences (epigenomics, genomics, proteomics, lipidomics, and transcriptomics) have provided important breakthroughs in understanding the pathophysiology of kidney diseases [5,6,7]. “The apogee” of omics research, known as metabolomics, seeks to elucidate and to provide a comprehensive view of the biochemical events that occur in cells, as well as the relationships between these processes in biological specimens. Hence, biomarker discovery in CKD has focused on metabolomic profiling. With a view to achieving a better understanding of biological systems, interest has shifted to a new powerful technology in order to discover potential diagnostic and therapeutic biomarkers. Emerging evidence has revealed that commonly applied techniques in metabolomic analysis are mass spectrometry (MS)-based techniques, including gas-chromatography/mass spectrometry (GC/MS), liquid chromatography/mass spectrometry (LC/MS), and magnetic resonance spectroscopy (MRS) [8].

Recent studies have shown that circulating levels of several metabolites (the tryptophane metabolic pathway, uremic toxins, amino acids, acylcarnitines, and various antioxidants) are altered by the kidney. Interestingly, the kidney has an important role in glomerular uptake, tubular secretion, and catabolism of the majority of metabolites. Several longitudinal studies have been applied for metabolomic profiling in CKD patients [9,10]. For example, a Korean study that followed up 1741 subjects for 8 years showed a positive correlation between kynurenine and kynurenine/tryptophane ratio and new-onset CKD. The authors also observed that several acylcarnitines, such as C3, C4, C7-DC, and C8, were associated with eGFR decline [9]. The Chronic Renal Insufficiency Cohort (CRIC) study is a prospect study that monitored CKD progression, defined as a 50% reduction in eGFR in a 6-year period, and included patients with eGFRs between 20 and 70 mL/min per 1.73 m^2^. This study measured both serum and urinary levels and found that 11 metabolites were associated with CKD progression. Therefore, it seems that kidney clearance of these metabolites can offer supplementary information in the assessment of kidney function [11].

Yuana D. et al., in a study performed on patients with end-stage renal disease (ESRD), defined as eGFR < 15 mL/min per 1.73 m^2^ and depression, were able to identify 57 metabolites from 19 metabolic pathways that were significantly different between ESRD patients with or without depression. In conclusion, they showed that oxidative stress, abnormal energy, and inflammation play a significant role in the development of depression in patients with ESRD [12].

Shima et al. showed that methionine, sulfoxide/methionine, and oxidative stress-related metabolites were associated with CKD [13]. Furthermore, metabolites seem to predict kidney dysfunction [14]; thus, a decrease in eGFR could alter the concentration of metabolites [15]. The evaluation of renal function is made with biomarkers, such as serum creatinine and blood urea, but these biomarkers have low specificity and sensitivity and they become relevant only in more advanced stages of CKD. Therefore, it is mandatory to discover more sensitive biomarkers for the early detection of kidney diseases.

The aim of the current study was to identify and characterize new potential blood and urine metabolomic biomarkers involved in the early diagnosis of CKD and to discover new therapeutic approaches using ultra-high-performance liquid chromatography coupled with electrospray ionization-quadrupole-time of flight-mass spectrometry.

## 2. Materials and Methods

### 2.1. Patients and Compliance with Ethical Standards

The protocol of the study, which included the study design, the collection of serum and urine samples, participant information, and written consent from all subjects enrolled, was approved by the Ethics Committee for Scientific Research of the “Victor Babeș” University of Medicine and Pharmacy Timișoara (no. 54/09.11.2020) and by the Ethics Committee of County Emergency Hospital Timișoara (no. 222/04.02.2021). A cohort of 88 non-diabetic CKD patients (group P), defined and staged according to the KDIGO Guideline for the Diagnosis and Management of CKD [16] and recruited form the Department of Nephrology, County Emergency Hospital Timișoara, and 20 healthy control subjects (group C) recruited from the general physicians’ records were included in the study from 1 February 2021 through to 31 July 2022. The CKD patients were divided into five groups based on the KDIGO CKD classification [13], as follows: group 1 (G1) included 15 patients with an eGFR of 90 mL/min/1.73 m^2^ or higher; group 2 (G2) included 15 patients with an eGFR of 89–60 mL/min/1.73 m^2^; group 3 (G3a) included 17 patients with an eGFR of 59–45 mL/min/1.73 m^2^; group 4 (G3b) included 15 patients with an eGFR of 44 to 30 mL/min/1.73 m^2^; group 5 (G4) included 15 patients with an eGFR of 29 to 15 mL/min/1.73 m^2^; and group 6 (G5) included 14 patients with an eGFR of less than 15 mL/min/1.73 m^2^. The exclusion criteria were diabetic kidney disease and the requirement of renal replacement therapies. Blood and urine samples were taken in the morning and after a 12 h fasting period. For both groups, additional clinical and biological measurements were simultaneously collected and registered (Table 1).

### 2.2. Sample Collection and Processing

Blood was collected by venepuncture in sterile vacutainers without anticoagulant, and the serum was stored at −80 °C until analysis. The first morning urine samples were collected in sterile vials. All samples were labelled using confidential numerical codes. A volume of 0.8 mL of a mix of pure HPLC-grade Methanol and Acetonitrile (2:1 *v*/*v*) was added for each volume of 0.2 mL of serum and 0.2 mL urine. In each case, the mixture was vortexed to precipitate proteins, ultrasonicated for 5 min, and kept for 24 h at −20 °C to increase the protein precipitation. The supernatant was collected after centrifugation at 12,500 rpm for 10 min (4 °C) and filtered through nylon filters (0.2 μm). Finally, the supernatant was placed in glass micro-vials and introduced into the autosampler of the ultra-high-performance liquid chromatograph (UHPLC) before injection.

### 2.3. UHPLC-QTOF-ESI+−MS Analysis

The metabolomic profiling was performed by ultra-high-performance liquid chromatography coupled with electrospray ionization-quadrupole-time of flight-mass spectrometry (UHPLC-QTOF-ESI+−MS) using a Thermo Fisher Scientific (Waltham, MA, USA) UHPLC Ultimate 3000 instrument equipped with a quaternary pump, a Dionex (Sunnyvale, CA, USA) delivery system, and MS detection equipment with MaXis Impact (Bruker Daltonics, Billerica, MA, USA). The metabolites were separated on an Acclaim C18 column (5 μm, 2.1 × 100 mm, pore size of 30 nm; Thermo Fisher Scientific, Waltham, MA, USA) at 28 °C. The mobile phase consisted of 0.1% formic acid in water (A) and 0.1% formic acid in acetonitrile (B). The elution time was set for 20 min. The flow rate was set at 0.3 mL·min^−1^ for serum samples and 0.8 mL·min^−1^ for urine samples. The gradient for serum samples was: 90 to 85% A (0–3 min), 85–50% A (3–6 min), 50–30% (6–8 min), 30–5% (8–12 min), and afterwards increased to 90% at min 20. The gradient for urine samples was: 90 to 85% A (0–3 min), 85–30% A (3–6 min), 30–10% (6–8 min), isocratic until min 12 and then increased to 90% at min 20. The volume of injected extract was 5 mL, and the column temperature was set at 25 °C. Several QC samples obtained from each group were used in parallel to calibrate the separations. Doxorubicin hydrochloride (m/z = 581.3209) solution (0.5 mg/mL) was added in parallel to QC samples as an internal standard.

The applied MS parameters were: ionization mode positive (ESI+), MS calibration with Natrium format, capillary voltage 3500 V, the pressure for the nebulizing gas set at 2.8 bar, drying gas flow of 12 L/min, and drying temperature of 300 °C. The m/z values to be separated were set between 60 and 600 Daltons. The control of the instrument and the data processing was performed using the specific software packages TofControl 3.2, HyStar 3.2, Data Analysis 4.2 (Bruker Daltonics, Billerica, MA, USA), and Chromeleon.

### 2.4. Statistical Analysis

The Bruker software Data Analysis 4.2, attached to the instrument, was used to process the acquired data. By using the peak dissect algorithm, details of the molecules separated were obtained. Using the algorithm Find Molecular Features (FMF), a first advanced bucket matrix was generated. This included the retention time, the peak area, the peak intensity, and the signal/noise (S/N) ratio for each m/z value. From the total ion chromatograms, using specific algorithms, the TICs (total ion chromatograms) and BPCs (base peak chromatograms) were obtained. The number of separated molecules (m/z values) ranged between 320 and 420 in serum samples and reached up to 550 in urine samples.

In a first step, the molecules with retention times below 0.8 min, the molecules with S/N values < 5, the molecules with m/z values over 480 Daltons (Da), and the minor molecules and residues with peak intensities under 1000 units were eliminated. The number of molecules selected for statistics decreased to 200–250.

In a second step, the alignment of common molecules (with the same m/z values) in all samples was performed, keeping for the final matrix the molecules common to more than 80% of samples. Therefore, in the final matrices, the numbers of common molecules (m/z values) from serum and urine were 130 and 194, respectively. The alignment was performed using online software available at: www.bioinformatica.isa.cnr.it/NEAPOLIS (accessed on 15 July 2022). Next, these molecules were introduced in the Metaboanalyst 5.0 platform (https://www.metaboanalyst.ca/, accessed on 15 July 2022) and multivariate and univariate analysis was performed.

The untargeted metabolomic analysis was performed by multivariate analysis comparing group C with the whole group P, based on the final matrices.csv for each type of sample (serum and urine). The discriminations between these 2 groups included fold changes, volcano tests, PatternHunter analysis, partial least squares discriminant analysis (PLSDA), sparse PLSDA (sPLSDA), and variable importance in the projection (VIP) values, including cross-validation parameters. Then, the random-forest-based prediction test was applied, and calculations of *p*-values were performed by *t*-tests. The heatmaps of correlations were also built. Finally, using the biomarker analysis, the receiver operating curves (ROCs) and the values of the areas under the ROC curves (AUCs) were obtained, and the m/z values were ranked according to sensitivity/specificity.

Univariate analysis allowed comparison of the subgroups G1–G5 with the control group. The statistical analysis was performed for each type of sample (serum and urine) using one-way ANOVA, PLSDA and sPLSDA score analyses, HunterPattern, random forests, and heatmaps.

The results were presented graphically, and the putative biomarkers of differentiation were identified. The identification of molecules, based on their m/z values and retention times, was performed in agreement with our database and other international databases for metabolomics: the Human Metabolome Database (http://www.hmdb.ca, accessed on 15 July 2022), Lipid Maps (http://www.lipidmaps.org, accessed on 15 July 2022), and PubChem (https://pubchem.ncbi.nlm.nih.gov, accessed on 15 July 2022).

## 3. Results

### 3.1. Multivariate Analysis of Serum Samples

#### 3.1.1. PLSDA Score Plot and VIP Scores

The homogeneity of the groups and the discrimination between groups C and P is presented in the PLSDA score plot (Figure 1a). With a covariance of 23.3%, the discrimination is significant and can be explained by the significance of the differences between some metabolites, the first 15 being mentioned in the VIP score plot (Figure 1b).

According to the PLSDA plot, a division of group P into two subgroups was observed—an aspect which suggested certain molecular features which may explain this discrimination. According to VIP scores (>1.8) and MDA values (>0.004), in group P, significantly decreased molecules with m/z values of 340.2809 (Oleoyl glycine), 207.1736 (Alpha-Lipoic acid), 171.1608 (Propylthiouracil), and 121.9741 (L-Cysteine) were observed, along with increased levels of molecules with m/z = 183.0079 (Sorbitol), 216.0103 (Propenoylcarnitine), 179.0162 (Cysteinylglycine), and 192.9925 (5-Hydroxyindoleacetic acid). Using the cross-validation algorithm, high accuracy (close to 1), high R² values (>0.5), and a significant Q2 value (>0.4) for the first four components was identified. Therefore, the model could be considered predictive.

#### 3.1.2. Biomarker Analysis and Prediction by Random Forest Analysis

According to Metaboanalyst software, the biomarker analysis included the representation of the receiver operating characteristic (ROC) curve and the area under the curve (AUC). Therefore, the sensitivity vs. specificity of each molecule identified as a potential biomarker was evaluated. In good agreement with all previous studies, the biomarker analysis confirmed that molecules which can be considered biomarker candidates are the ones with the highest AUC values (>0.85). Table 2 shows the AUC value, *p*-value, and log2FC value for each molecule identified and the variations between group P and group C. Table 2 summarizes the data obtained and lists the most significant molecules to be considered, according to VIP and MDA values. The identification of these molecules was made considering their m/z values in the HMDB database.

These data confirm that Oleoyl glycine, Alpha-Lipoic acid, All-trans retinoic acid, Sorbitol L-Cysteine, Propenoylcarnitine, and Kynurenic acid can be considered potential biomarkers for a significant discrimination between group C and group P.

### 3.2. Univariate Analysis of Serum Samples

#### One-Way ANOVA Applied for the Identification of Biomarkers of CKD Progression (G1–G5)

By one-way ANOVA analysis, the significant molecules which may explain the progression of CKD, according to eGFR, as seen in differences between the subgroups G1–G5 and group C, were identified.

The classification and discrimination of blood molecules was made with the PLSDA score plot (Figure 2a), followed by the VIP score (Figure 2b) and random forest score plots (Figure 2b). The PLSDA score plot had a covariance of 20.1% that was able to discriminate between the CKD subgroups G1–G5 and group C. The VIP scores show the ranking of the first 15 molecules to be considered responsible for the discrimination, as presented in Figure 2b. The cross-validation plot shows the low accuracy of the results: R2 values were higher than 0.1, but Q2 values were <0 for all components. By random forest analysis, we identified 15 molecules as potential biomarkers for CKD subgroups (Figure 2c).

In agreement with the VIP and MDA scores (Figure 2b,c), the most significant molecules to be considered predictive for the discrimination between group C and subgroups G1–G5 and with progressive decrease in levels from group C to the groups with lower eGFRs, namely, subgroups (G1 to G5), were Oleoyl glycine (340.2809), N-Butyrylglycine (m/z = 144.9693), Propylthiouracil (m/z = 171.1608), Glutamine (m/z = 147.1197), and Ketoleucine (m/z = 131.1162). The opposite ranking shows molecules with increased levels in subgroups G1 to G5 as compared to group C, e.g., 5-Hydroxyindoleacetic acid (m/z = 192.9925), Phenylalanine (m/z = 166.0979) Pyridoxamine (m/z = 169.1529), Cysteinylglycine (m/z = 179.0152), Propenoylcarnitine (m/z = 216.0103), and Uridine (m/z = 245.0952).

### 3.3. Multivariate Analysis of the Urine Samples

#### 3.3.1. PLSDA Score Plot and VIP Scores

For urine samples, the PLSDA score plot (Figure 3a) shows a covariance of 20.1% and a good discrimination between groups C and P. Meanwhile, according to the VIP scores (Figure 3b), the first 15 molecules ranked were considered responsible for the discrimination. The cross-validation plot showed an acceptable accuracy level for the results: R2 values were higher than 0.5, but Q2 values were <0 for all components.

According to the VIP scores, in group P, increased levels for almost all molecules (excepting Gluconolactone with m/z = 180.0697) were observed, the most significant (VIP > 2.4) being the molecules with m/z values of 214.2524 (Indoxyl sulfate), 235.1712 (Methoxytryptophan), 301.1441 (All-trans retinoic acid), 329.0086 (Glycylprolylarginine), 275. 1642 (Serotonin sulfate), and 279.1616 (Leucyl-phenylalanine).

#### 3.3.2. Biomarker Analysis and Prediction by Random Forest Analysis

The biomarker analysis included the representation of the ROC curve and AUC values. The most significant biomarker candidates, the ones with the highest AUC values (>0.900) and with higher MDA values, according to random forest analysis, were proposed. Table 3 shows the AUC, VIP, and MDA values, as well as the *p*-value and log2FC value for each molecule identified and variations between group P and group C. The identification of these molecules was made considering their m/z values in the HMDB database.

### 3.4. Univariate Analysis of Urine Samples

The same steps were performed for the urine samples: one-way ANOVA was used to determine significant molecules which may explain the differences between the subgroups G1–G5 and group C in urine samples.

Urinary molecules were categorized and distinguished using the PLSDA scores (Figure 4a), which had a PLSDA covariance of 19.8%. Figure 4b presents the ranking of the first 15 molecules proposed as representative for the discrimination considering the VIP scores. By random forest analysis, we identified 15 molecules as potential biomarkers for CKD subgroups G1–G5 (Figure 4c).

According to the VIP values above 2.0 and MDA values above 0.007, almost all molecules showed increased levels in groups G4–G5 (having lower eGFR levels) as compared to controls and groups G1–G3 with higher eGFR levels. The most significant molecules were Indoxyl sulfate (m/z = 214.2544), All-trans retinoic acid (m/z = 301.1441), Glycylprolylarginine (m/z = 329.0086), Leucyl-phenylalanine (279.1616), Methoxytryptophan (m/z = 235.1712), Methylarachidic acid (m/z = 327.0114), Serotonin sulfate (m/z = 275.1642), and Butenoylcarnitine (m/z = 230.2496).

In order to integrate the data obtained by non-targeted analysis, the selection of the most representative molecules considered as potential biomarkers in serum vs. urine is presented in Table 4. Considered were the data released by multivariate analysis (P vs. C, columns 2 and 4), as well the trends of such molecules, which are dependent on eGFR (decreased values from subgroup G1 to G5, respectively) (columns 3 and 5).

According to the data presented above (multivariate and univariate analysis by different algorithms), the selection of the molecules was made based on their statistical significance (*p* < 0.05) and correlated with data from the literature. The classification of these molecules was made using their retention times, m/z values, and peak intensities.

## 4. Discussion

The results of this study provide an insight into various metabolic pathways involved in the pathogenesis of CKD. The metabolites identified in CKD patients may impact both glomerular and tubular structures, even in the early stages of CKD. The findings of this study show that patients with CKD display a specific metabolomic profile, which may serve as an indicator of CKD initiation and progression.

### 4.1. Molecules to Be Considered as Potential Biomarkers Which Allow the Identification of Early Stages of CKD

Our data complete the existing reports and underline specific categories of molecules involved in the pathogenesis of CKD, such as amino acids (e.g., phenylalanine and L-cysteine), acylated amino acids (oleoyl glycine), and derivatives of tryptophan (methoxytryptophan and kynurenic acid), peptides (glycylprolyl arginine and leucylphenylalanine), sorbitol, and all-trans retinoic acid.

Specifically in urine, increases in indoxyl sulfate and in serotonin as serotonin sulfate can be considered significant signals for the early diagnosis of CKD and its progression from high to low eGFR. Therefore, we consider these metabolites as prognostic candidates for CKD diagnosis and progression monitoring.

#### 4.1.1. Amino Acids

Amino acids (AAs) regulate proteolysis and hemodynamics to maintain the integrity of the kidney. In 99 percent of cases, amino acids are filtered by the kidney and reabsorbed by the renal tubule. In our study, we observed decreased serum levels for serine, taurine, cysteine, ornithine, tyrosine, and L-tryptophan and increased serum levels for proline, valine, threonine, and phenylalanine. Regarding urinary levels of AA metabolites, we observed that these were significantly increased as compared to their plasma levels. Garibotto et al. showed that AA levels are significantly altered in both plasma and urine in CKD patients, with significantly lower plasma levels vs. urinary levels [17].

Phenylalanine, an essential amino acid that is not normally synthesized by the human body, decreases progressively in CKD [18]. Phenylalanine hydroxylase, located in the kidney, liver, and pancreas, also decreases in CKD.

In the present study, we found that phenylalanine correlated positively with eGFR. These findings are in contradiction with several previous studies that showed increased plasma levels of phenylalanine in patients with CKD and IgA nephropathy. Psihogios et al. [18], in a study performed on patients with glomerulonephritis, found decreased urine levels of phenylalanine [19].

Interestingly, we observed that circulating levels of phenylalanine were higher in patients classified as having CKD with normal eGFRs compared with group G4. Li et al. also found that serum levels of phenylalanine were slightly increased in patients with CKD but normal kidney function and that levels continued to increase as CKD progressed [20]. Based on these observations, we believe that phenylalanine can be used as an early biomarker for CKD, though further studies are needed.

Cysteine (CysSSP) is a non-essential amino acid that can be acquired from food or by methionine cleavage. CysSSP is involved in protein synthesis, protein structural stabilization, glutathione formation, and the synthesis of intracellular metabolites and signaling molecules (e.g., taurine, coenzyme A, hydrogen sulfide, and cysteine persulfate) [21]. However, cysteine appears to play a part in metabolic syndrome, obesity, and insulin-like effects on adipocytes. By producing reactive oxygen species and reducing endothelial vasodilatation, it has been linked to inflammation and endothelial dysfunction.

Our data showed that Cysteine levels were increased in the plasma of patients classified as G1 and G2, as compared with group C and sub-groups G3a and G3b. Similar results were also observed in the patients’ urine.

Sumayao et al. provided insights into the distribution of significant amounts of cysteine, glutathione, and cysteine disulfides along the proximal tubule and pointed to the fact that lysosomal uptake serves as the mechanism for reabsorption [22]. While simultaneously acting as a secure method for cysteine storage, high levels of CysSSP can cause protein misfolding and endoplasmic reticulum stress [23]. In a study performed on adult rats, it was shown that plasma levels of cysteine were controlled by the liver–kidney axis and that the kidney cysteine-to-glutathione ratio grew four times as compared to two to three times in the liver, thus demonstrating that either the kidney or the liver utilize and store cysteine. A change in senile kidney ability to absorb cysteine seems to be the cause of elevated plasma levels of CysSSP in CKD patients [24].

Interestingly, our study found a correlation between blood levels of cysteine and renal function: the group of patients classified as having G3b had the lower ratio, and the concentration in the urine rose with the decline in eGFR, indicating that the tubular cells were unable to reabsorb and store cysteine.

5-methoxytryptohan (5-MTP) is an endogenous molecule synthetized from tryptophane breakdown. Among other metabolites, 5-MTP was one of the metabolites that was strongly correlated with eGFR and kidney disease progression, these data being consistent with previous studies [25]. Furthermore, the anti-inflammatory [26] and protective effects of 5-MTP in vascular injuries associated with CKD were demonstrated by several studies [27]. In our study, 5-MTP levels were decreased in serum and increased in the urine in patients with CKD compared with the control group. These findings are supported by other studies performed on patients with CKD and by in vivo studies performed on mice with unilateral ureteral obstruction. Interestingly, several studies showed that supplementation with 5-MTP can ameliorate tissue injury, fibrosis, and inflammation [25].

Furthermore, 5-MTP can be used as an early biomarker for CKD and also as a potential treatment for CKD progression by attenuating inflammation and kidney fibrosis. Therefore, 5-MTP can be considered to be the leading compound for the development of novel anti-fibrotic drugs.

Based on these observations, we may conclude that increased urinary levels and decreased serum levels of AAs in CKD patients are explained by the fact that the Na+−coupled reabsorption of AAs by the proximal tubule is altered. These findings are supported also by the fact that, as kidney dysfunction progresses, urinary levels of AAs increase. Thus, L-Cysteine concentration in the urine increased gradually with the decline in eGFR. AA metabolites could be used as serum and urinary biomarkers for early detection and monitoring of the progression of CKD.

#### 4.1.2. Acylcarnitines

A specific class of molecules reported to be significant in CKD are the acylated carnitines involved in mitochondrial transport of lipids. Acylcarnitines are involved in lipid metabolism, inflammation, amino acid metabolism, and mitochondrial activity [28]. The function of long-chain acylcarnitines is to carry fatty acids through the mitochondrial inner membrane for beta-oxidation. Therefore, they are considered intermediate products of beta-oxidation. In contrast, the role of short- and medium-chain acylcarnitines appears to be connected to the metabolism of amino acids [29]. Since the glomerulus is primarily responsible for acylcarnitine excretion, decreased kidney function is linked to both an increase in the blood levels of acylcarnitine and a decrease in its excretion [30]. Long-chain acylcarnitines are the first products of β-oxidation and, as such, mitochondrial dysfunction due to lipotoxicity may contribute to the formation of long-chain acylcarnitines and the development of kidney dysfunction. Acylcarnitines have been linked to a higher risk of developing diabetes mellitus and have been shown to be able to predict the development of cardiovascular disease [31].

Our research demonstrated the significance of acylcarnitines as a group of key metabolites in kidney disease. Moreover, we observed a significant correlation between eGFR and plasma and urinary levels of short- and medium-chain acylcarnitines (L-carnitine, L-Acetylcarnitine, Propionylcarnitine, and Butenylcarnitine). The most significant were identified as propenoylcarnitine and butenoylcarnitine, the serum and urine levels of which were increased in the CKD group of patients as compared with the C group.

Our findings are supported by other studies which showed that short- and medium-chain acylcarnitines (C3-propinylcarnitine, C4-butyrylcarnitine, C7-DC-Pimelylcarnitine, and C8-Octanoylcarnitine) were linked to decline in eGFR [9]. Mitochondrial dysfunction is correlated with the development of various diseases. Therefore, blood levels of acylcarnitines could provide a better understanding of how alterations in mitochondrial metabolism are involved in the progression, manifestation, and severity of different diseases, including CKD.

We assume that the accumulation of acylcarnitines in plasma is due to glomerular dysfunction. This observation is supported by the fact that the levels of L-carnitine, L-Acetylcarnitine, and Propionylcarnitine in serum increased gradually as renal function declined. In summary, we showed that acylcarnitines are good candidates as plasma and urinary biomarkers for the early diagnosis of CKD, though additional targeted metabolomic studies are required for validation.

#### 4.1.3. Uremic Toxins

In the past decade, a number of uremic molecules have been categorized and identified and their functions in the onset and course of CKD and its complications have been established. Protein-bound uremic toxins are of gastrointestinal origin due to the ability of the intestinal flora to break down aromatic acids [32].

In our study, the uremic toxins identified included indoxyl sulfate, xanthine, hippuric acid, and uridine, which data are consistent with other studies regarding the numbers of uremic toxins that occur in patients with renal dysfunction.

Production of hippuric acid begins in the gut when dietary polyphenols are converted by the gut microbiota into benzoic acid, and it is completed in the liver or kidney when glycine is conjugated to hippuric acid [33]. In our study, increased serum levels of hippuric acid were observed in patients with CKD vs. group C, with higher levels in urine vs. plasma. Univariate analysis showed that patients in groups G1 and G2 had higher serum levels compared with groups C, G3a, and G3b. Regarding urinary levels, we observed that patients in group G1 and group G3b had increased excretion of hippuric acid compared to group G2 and group G3b.

Hippuric acid is a protein-bound uremic toxin that is bound to albumin and correlates with disease progression in CKD patients. The full mechanisms of renal toxicity have not been characterized, despite several findings showing the role of hippuric acid in the progression of renal fibrosis and induction of oxidative stress and accumulation of ROS, with subsequent endothelial dysfunction. Moreover, according to a recent study, hippuric acid can disturb redox equilibrium and increase the expression of several genes which have been linked to the development of renal fibrosis and extracellular matrix imbalance [33].

Presumably due to the fact that hippuric acid is a protein-bound uremic toxin, we believe that increased urinary levels of hippuric acid in CKD patients occur both in relation to increased protein loss and altered glomerular filtration. In addition, we assume that these patients have increased renal fibrosis and significant proximal tubule dysfunction, thus allowing large amounts of hippuric acid to be secreted into the urine. These findings suggest that hippuric acid can be used as a urinary biomarker for the assessment of CKD progression.

In the intestinal flora, tryptophane is broken down into indole by intestinal bacteria, which begins the synthesis of indoxyl sulfate (IS). Next, cytochrome P450 2E1 uses liver sulfotransferase 1A1 to hydroxylate indole, resulting in 3-hydroxy indole. The end product is indole sulfate, which is created when 3-hyrosxy indole is sulfonated [34].

In our research, plasma and urinary levels of IS were increased in CKD patients vs. controls, being more increased in urine vs. serum, respectively. Furthermore, by performing univariate analysis we observed that patients in groups G1 and G2 had higher serum levels as compared with group C and groups G3a and G3b, respectively. Of note, we found increased levels of IS even in the early stages of CKD.

The nephrotoxic consequences of IS are represented by depletion of antioxidants, generation of reactive oxygen species, promotion of fibrosis, and inflammation. IS can affect renal function and cause proteinuria and podocyte dysfunction through the activation of aryl hydrocarbon receptors in the glomerulus, which downregulates a number of proteins important in maintaining cell integrity [34]. Basolateral anion transporters 1 and 3 in the proximal tubule are responsible for IS excretion [35]. In proximal tubule cells, IS can activate NF-kB, thus suppressing cellular proliferation and stimulating production of PAI-1 and promoting TGF-β1-induced fibrosis [36]. In addition, IS can induce the development of intracellular adhesion molecule-1, which enhances monocyte infiltration into the kidney and monocyte chemoattractant protein, which increases the recruitment of macrophages and causes tubulointerstitial inflammation [37].

It has been shown by Fujii et al. that decreased renal function is associated with higher levels of IS [38]. Dialysis has a minimal impact on IS plasma levels because it is a protein-bound uremic toxin [39].

In our research, we found higher blood levels of IS in patients with early-stage CKD as compared to controls, and higher urine levels were associated with a decline in renal function. As a result, we assume that there are two main reasons that can explain these findings. On the one hand, IS is highly bound to albumin, and therefore glomerular impairment leads to increased filtration, while, on the other hand, proximal tubule dysfunction can result in increased excretion. Moreover, we found that uremic toxins, such as IS and hippuric acid, may be used as plasma biomarkers for CKD.

Sorbitol is a component of the tricarboxylic acid (TCA) cycle and a carbohydrate-related metabolite that can be produced naturally or synthetically from glucose. Previous studies showed that sorbitol is a molecule considered as a potential biomarker in diabetic nephropathy, where the elevated extracellular concentration of glucose disturbs cellular osmoregulation and sorbitol is synthesized intracellularly via the polyol pathway and is also considered a potential uremic toxin [40]. In this study, we found that circulating levels of sorbitol were directly correlated with eGFR. These findings are in disagreement with those of Roshanravan et al., who found that sorbitol levels were decreased in patients with CKD. It is well documented that sorbitol accumulates intracellularly, inducing osmotic swelling, as well as in the renal cortex in hyperglycemic states [38,41]. CKD is characterized by insulin resistance/altered glucose metabolism, and this seems to play a significant role in the development and progression of pathological changes in the kidney, the main compartments affected being vascular and tubulointerstitial. Due to the fact that glucose reabsorption and concentration happens in the proximal tubule, it can be considered the most predisposed segment of the renal tubules to injury. Ishii et al. showed that prescribing aldose reductase inhibitors can decrease tissue damage as well as the excretion of different urinary enzymes [42]. Interestingly, we found that circulating levels of sorbitol increased gradually from group C to group G5. Given these findings, we believe that sorbitol accumulation occurs also in non-diabetic CKD patients, the main reason being that CKD is characterized by altered glucose metabolism and proximal tubular dysfunction.

#### 4.1.4. Antioxidants

Alpha-Lipoic acid (ALA), also known as thioctic acid, is a compound that contains sulfur and can be found in plants, animals, and humans. ALA is also a vitamin-like antioxidant which was reported recently to have an impact on CKD pathogenesis [40,43]. In this study, we observed that serum levels of ALA correlated negatively with eGFR. ALA has been used in in vitro and in vivo studies [44,45,46]. It has been proven that ALA is a strong antioxidant that can increase vitamin E, vitamin C, catalase, and glutathione activity, and it can also act as a free radical scavenger and a metal chelator and restore oxidative injury and antioxidant defense [46,47]. Other relevant benefits of ALA supplementations are related to its anti-inflammatory actions, improving endothelial nitric oxide syntheses by acting on different signaling cells, and contribution to metabolic pathways correlated with mitochondria [48]. Therefore, this metabolite has become useful in the management of several vascular diseases and diabetic complications, such as retinopathy and neuropathy. Zang et al. [49] showed that ALA supplementation can be useful in acute kidney prevention [50]. Takaoka et al. proved that supplementation with ALA can also prevent glomerular injury due to diabetes mellitus [51]. In addition, other studies performed on mice showed that ALA supplementation can prevent toxic injuries induced by methotrexate and cisplatin in the kidney [52] and that mice with unilateral obstruction had minimal fibrosis and moderate histological renal damage [53].

As mentioned above, we observed that circulating levels of ALA decreased gradually in group C and group G1 to group G5. We assume that a correlation between impaired nutrition in patients with advanced CKD and the fact that ALA has a short life and decreased bioavailability due to hepatic degradation, decreased solubility, and instability in the gastro-intestinal system can explain the progressive decrease in circulating levels of ALA as kidney function declines. ALA levels are age-dependent, and therefore this observation could explain the fact that ALA levels are more decreased in CKD. Another plausible explanation can be offered by the fact that in humans ALA is found in mitochondria and that CKD is characterized by mitochondrial dysfunction from its early stages. We believe that lipoic acid supplementation as a natural antioxidant could be used as a therapeutic approach for CKD progression.

Our study has several limitations. First, this is a cross-sectional study which does not allow the establishment of relations of causality between phenomena. Second, the small sample size and the heterogeneity with regard to the causes of CKD could have induced bias in the interpretation of data and decreased the statistical power of the study.

The strengths of our study reside in the identification of a group of metabolites belonging to various metabolic pathways related to CKD pathogenesis. Additionally, the metabolites found in serum and urine displayed a particular behavior according to CKD stage, allowing for the characterization of a specific metabolic profile of patients even in the early stages of CKD.

## 5. Conclusions

In conclusion, in our study, high levels of acylcarnitines, antioxidants, tryptophane metabolites, uremic toxins, and amino acids have been found in all CKD stages. Their dual variations in serum and urine may explain their impact on both glomerular and tubular structures, even in the early stages of CKD. The particular metabolomic profile found in our study could label the metabolites identified as potential biomarkers useful in the diagnosis of early CKD. Further longitudinal studies applying targeted metabolomic analyses of blood and urine metabolites are required in order to establish relations of causality between these metabolites and CKD progression.

## Figures and Tables

**Figure 1 biomedicines-11-01057-f001:**
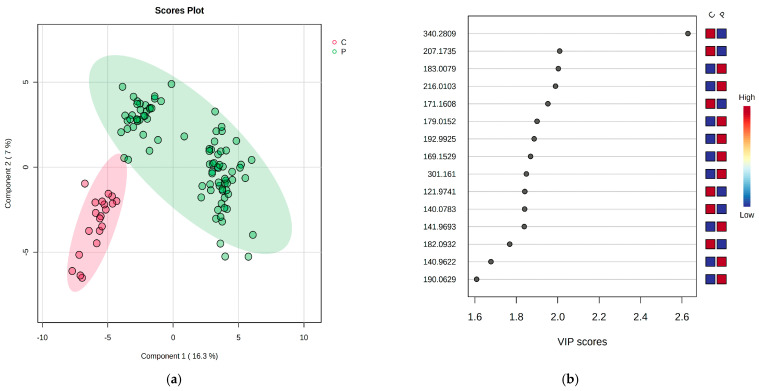
(**a**) The PLSDA score plot highlights the good discriminations between the C group and the CKD (P) group. (**b**) VIP values and ranking of top 15 molecules according to PLSDA analysis.

**Figure 2 biomedicines-11-01057-f002:**
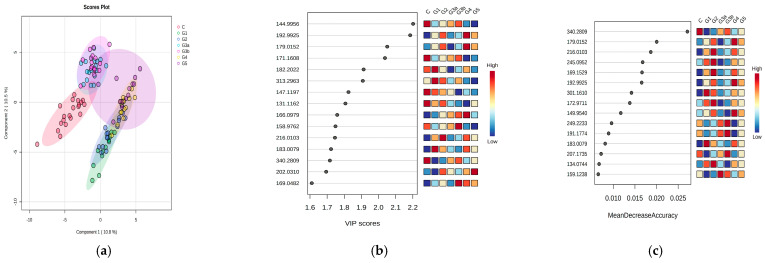
(**a**) PLSDA score plot representing the discrimination between group C and subgroups G1–G5 in blood serum. (**b**) VIP score plot for the top 15 molecules, with VIPs > 1.6. (**c**) MDA values for the top 15 molecules to be considered as potential biomarkers of progression from G1 to G5.

**Figure 3 biomedicines-11-01057-f003:**
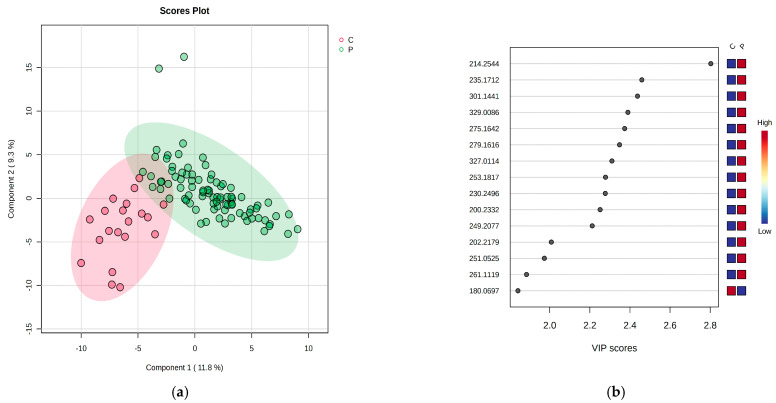
(**a**) The PLSDA score plot highlights the good discriminations between the C group and group P for urine samples. (**b**) VIP values and ranking of top 15 molecules according to PLSDA analysis.

**Figure 4 biomedicines-11-01057-f004:**
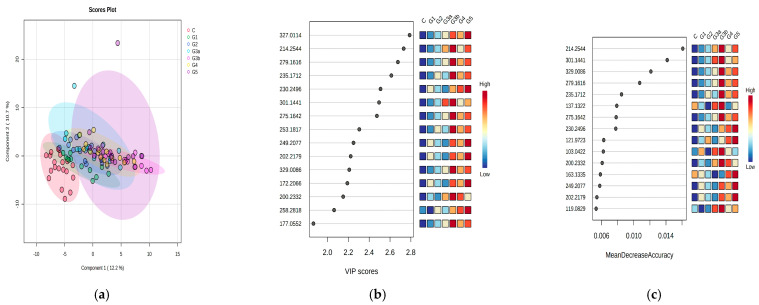
(**a**) PLSDA score plot representing the discrimination between group C and subgroups G1–G5 in urine samples (covariance of 19.8%). (**b**) VIP score plot for the top 15 molecules, with VIPs > 1.6. (**c**) MDA values for the top 15 molecules to be considered as potential biomarkers of progression from G1 to G5.

**Table 1 biomedicines-11-01057-t001:** Demographic, clinical, and biological data for group C and the CKD groups.

		P Group					
	C Group	G1 Group	G2 Group	G3a Group	G3b Group	G4 Group	G5 Group
Participants	20	12	15	17	15	15	14
Sex (M)	12	7	9	6	7	8	6
Age (y)	55.85 ± 7.25	39.92 ± 10.8	53.6 ± 15.4	55.1 ± 15.2	58.9 ± 14.4	61.2 ± 14.8	63.6 ± 12.6
BMI (kg/m^2^)	25.35 ± 8.5	26.42 ± 3.1	26.9 ± 1.7	27.9 ± 1.9	28.5 ± 2.3	27.3 ± 3.2	28.6 ± 2.2
Glomerulonephritis	0	3	5	1	0	5	3
Hypertension	0	12	15	17	15	15	14
Acquired solitary kidney	0	0	0	0	0	2	1
Serum creatinine (mg/dL)	0.73 ± 0.08	1.46 ± 2.1	1.4 ± 0.3	1.6 ± 0.7	1.7 ± 0.3	6.5 ± 13	5.19 ± 0.8
eGFR(ml/min/1.73 m^2^)	97.93 ± 11.71	101.9 ± 12.2	65.1 ± 12.9	49 ± 10.6	39.9 ± 5	21.3 ± 11.6	12.9 ± 11.8
uACR (mg/g)	14.67 ± 6.4	449.7 + 1177.3	1252.3 + 1625.7	630.6 + 709.2	672.9 + 1509.1	747.9 + 884	1102.9 + 1365.6

Legend: M: male; BMI: body mass index; e-GFR: estimated glomerular filtration rate; uACR: urine albumin-creatinine ratio.

**Table 2 biomedicines-11-01057-t002:** The m/z values for the most significant metabolites from serum samples and their identification according to the HMDB database. These metabolites were considered predictive given their VIP and MDA values.

m/z	Identification	HMDB ID		VIP	MDA	AUC	*p*-Value	Log2 FC
340.2809	Oleoyl glycine	HMDB0013631	D	2.629	0.016	0.961	6.59 × 10^−14^	0.773
207.1735	Alpha-Lipoic acid	HMDB0001451	D	2.009	0.017	0.904	6.57 × 10^−8^	
141.9693	Ethanolamine Phosphate	HMDB0000224	I	1.838	0.009	0.903	1.07 × 10^−6^	−2.113
301.161	All-trans retinoic acid	HMDB0001852	I	1.849	0.008	0.907	9.16 × 10^−7^	−2.133
183.0079	Sorbitol	HMDB0000247	I	2.003	0.007	0.891	7.57 × 10^−8^	−1.146
121.9741	L-Cysteine	HMDB0000574	D	1.841	0.007	0.851	1.03 × 10^−6^	0.363
216.0103	Propenoy lcarnitine	HMDB0013124	I	1.989	0.006	0.869	9.55 × 10^−8^	−1.104
166.0979	Phenylalanine	HMDB0000159	D	1.840	0.005	0.828	7.39 × 10^−5^	−0.490
190.0629	Kynurenic acid	HMDB0000715	I	1.608	0.004	0.831	2.57 × 10^−5^	−0.628

**Table 3 biomedicines-11-01057-t003:** The m/z values for the most significant metabolites from urine samples and their identification according to the HMDB database. These metabolites were considered predictive given their VIP and MDA values.

m/z	Identification	HMDB ID		VIP	MDA	AUC	*p*-Value	Log2 FC
329.0086	Glycylprolylarginine	HMDB0252828	I	2.409	0.009	0.955	5.53 × 10^−9^	−0.758
253.1817	Deoxyinosine	HMDB0000071	I	2.338	0.006	0.932	3.52 × 10^−8^	−0.56
279.1616	Leucyl-phenylalanine	HMDB0013243	I	2.389	0.006	0.92	1.12 × 10^−8^	−0.771
230.2496	Butenoylcarnitine	HMDB0249460	I	2.283	0.006	0.919	3.58 × 10^−8^	−0.844
301.1441	All-trans retinoic acid	HMDB0001852	I	2.484	0.008	0.918	2.39 × 10^−9^	−1.091
235.1712	Methoxy tryptophan	HMDB0002339	I	2.508	0.008	0.918	1.62 × 10^−9^	−1.114
275.1642	Serotonin sulfate	HMDB0240717	I	2.380	0.002	0.907	7.30 × 10^−9^	−0.754
214.2544	Indoxyl sulfate	HMDB0000682	I	2.823	0.014	0.968	1.34 × 10^−2^	−1.128

**Table 4 biomedicines-11-01057-t004:** Serum and urine metabolites selected by untargeted analysis as potential biomarkers for the discrimination between group P and group C, considering the data released from multivariate analysis (P vs. C, columns 2 and 4), as well the trends of such molecules dependent on eGFR (decreased values from G1 to G5 subgroup, respectively; columns 3 and 5).

Serum Metabolites	P vs. C	Subgroups vs. Controls	Urine Metabolites	P vs. C	Subgroups vs. Controls
Oleoyl glycine	D	G5 < G1 < C	Indoxyl sufate	I	G5 > G1 > C
Alpha-Lipoic acid	D	G5 < G1 < C	Glycylproly lArginine	I	G5 < G1 < C
All-trans retinoic acid	I	G5 > G1	Deoxyinosine	I	G3 > G1
Sorbitol	I	G5 > G1	Leucyl-phenylalanine	I	G5~G3 > G1 > C
L-Cysteine	D	G5 < G1 < C	Butenoylcarnitine	I	G5 > G1 > C
Propenoylcarnitine	I	G4~G2 > G1 > C	All-trans retinoic acid	I	G5~G3 > G1 > C
Phenylalanine	D	G4 < G1 < C	Methoxy tryptophan	I	G5~G3 > G1 > C
Kynurenic acid	I	G5 > G1 > C	Serotonin sulfate	I	G5~G3 > G1 > C

## Data Availability

The data that support the findings of this study are available from the corresponding author upon reasonable request.

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
