# Peer review of "Untargeted Metabolomics by Ultra-High-Performance Liquid Chromatography Coupled with Electrospray Ionization-Quadrupole-Time of Flight-Mass Spectrometry Analysis Identifies a Specific Metabolomic Profile in Patients with Early Chronic Kidney Disease"

_biomedicines, 2023, doi:10.3390/biomedicines11041057_

Round 1

Reviewer 1 Report

In the current paper by Glavan et al., it is reported the potential use of metabolites that are found in different CKD stages. The authors studied the samples of grouped CKD patients with ultra-high-performance liquid chromatography and electrospray ionization-quadrupole-time of flight-mass spectrometry. They described in detail their protocol for the study. According to this, the authors reported significant changes in urine and serum biomarkers since the early CKD stages, having the potential to be used in clinical practice to start prompt therapies during the disease. Below are some comments that need to be answered.

COMMENTS:

1)    Line 101, please mention the stage of CKD and if the study was acute or chronic.

2)    In line 103, please explain/mention if current biomarkers are ineffective for early CKD detection. 

3)    In the study protocol, why did the study choose the 17-month range for the follow-up of study controls?

4)    In Table 1, please accommodate the data better since some values overlap with others.

5)    Figure 1 appears before it is mentioned. Please pass below the result´s explanation.

6)    In line 246, there is an additional “to.”

7)    In the titles of the tables, please add if they are in urine or serum. It would provide more clarity.

8)    Please verify lines 398-400. Are long-chain acylcarnitines the first products of b-oxidation, or are they intermediate products? 

Author Response

Dear Editor-in-Chief and reviewers,

We want to start by thanking all for your suggestions, which were useful in helping us improve the manuscript. We took into consideration the Editor's and Reviewers’ comments and revised the manuscript. In the next lines, we will offer an answer to the specific comments being highlighted in red in the MS.

Please find attached our manuscript entitled “Untargeted metabolomic by ultra-high-performance liquid chromatography coupled with electrospray ionization-quadrupole-time of flight-mass spectrometry analysis identifies a specific metabolomic profile in patients with early chronic kidney disease” by Glavan Mihaela-Roxana et. al.           

REVIEWER 1

In the current paper by Glavan et al., it is reported the potential use of metabolites that are found in different CKD stages. The authors studied the samples of grouped CKD patients with ultra-high-performance liquid chromatography and electrospray ionization-quadrupole-time of flight-mass spectrometry. They described in detail their protocol for the study. According to this, the authors reported significant changes in urine and serum biomarkers since the early CKD stages, having the potential to be used in clinical practice to start prompt therapies during the disease. Below are some comments that need to be answered.

COMMENTS:

1)    Line 101, please mention the stage of CKD and if the study was acute or chronic.

- Thank you for pointing this issue to us. We decided to remove the sentence form our paper due to the fact that the reference was from 2003 and one of the main suggestions for this paragraph was to add studies published in the last 5 years. Nevertheless, to answer your question, the study was performed on chronic patients.

2)    In line 103, please explain/mention if current biomarkers are ineffective for early CKD detection.

- Thank you for your observation. We now mentioned in line 103 that the evaluation of renal function is made with biomarkers such as serum creatinine and blood urea, but these biomarkers have low specificity and sensitivity and they become relevant only in more advanced stages of CKD. Therefore, it is mandatory to discover more sensitive biomarkers for early detection of kidney diseases.

3)    In the study protocol, why did the study choose the 17-month range for the follow-up of study controls?

- Thank you for this useful recommendation. We choose the 17-month range for the follow-up of the study controls due to the fact that controls included in this study were recruited from the general physicians’ records. The majority of people screened were last time evaluated 1 to 2 years before enrolment in the study. Serum and urine analysis were performed to exclude those with unknown illnesses.

4)    In Table 1, please accommodate the data better since some values overlap with others.

- Thank you for pointing this issue to us.  We tried to adjust the data from Table 1 so that it does not overlap anymore. Please let us know if the situation persists.

5)    Figure 1 appears before it is mentioned. Please pass below the result´s explanation.

- Thank you for pointing this issue to us. We have changed the place of figure 1 and now is below the explanation of results.

6)    In line 246, there is an additional “to.”

   -Thank you for pointing this issue to us. We have removed the additional “to” from the line 246

7)    In the titles of the tables, please add if they are in urine or serum. It would provide more clarity.

- Thank you for this useful recommendation. We added in the titles of Table 2 that the metabolites presented are form serum and in Table 3 we specified that the results presented are form urine samples.

8)    Please verify lines 398-400. Are long-chain acylcarnitines the first products of b-oxidation, or are they intermediate products?

- Thank you for this pertinent observation. We have added at line 398-400 an explanation which states that long-chain acylcarnitines are intermediate products of fatty acid oxidation. They are widely used and produced in cellular energy metabolism pathways. Acylcarnitines have an important biological function, they are able to transport acyl groups form the cytosol to the mitochondrial matrix for β-oxidation and secondary to produce energy.

Reviewer 2 Report

Major revision is required.

1. Novelty for this study: Yes. The topic of metabolomic profile in patients with chronic kidney disease and diabetic kidney disease is emerging in the PubMed in recent years.

2. Major revision:

(1) Abstract: This study is a pilot study. Thus, the conclusion should be made conservatively.

(2) Introduction: The references cited in the fourth paragraph were older. There are more papers published in the PubMed in recent 5 years.

(3) Methods: This section of 2.4. Statistical analysis should be described in details. What are the goals and full names of PLSDA, VIP, MDA, log2FC, and HMDB? Which values were considered statistically significant?

(4) Figures 1-4: too small and unclear to read!

(5) References: less references in recent 5 years

Author Response

Dear Editor-in-Chief and reviewers,

We want to start by thanking all for your suggestions, which were useful in helping us improve the manuscript. We took into consideration the Editor's and Reviewers’ comments and revised the manuscript. In the next lines, we will offer an answer to the specific comments being highlighted in red in the MS.

Please find attached our manuscript entitled “Untargeted metabolomic by ultra-high-performance liquid chromatography coupled with electrospray ionization-quadrupole-time of flight-mass spectrometry analysis identifies a specific metabolomic profile in patients with early chronic kidney disease” by Glavan Mihaela-Roxana et. al.           

REVIEWER 2

Comments and Suggestions for Authors

Major revision is required.

  1. Novelty for this study: Yes. The topic of metabolomic profile in patients with chronic kidney disease and diabetic kidney disease is emerging in the PubMed in recent years.

- Thank you for raising this pertinent concern. We analysed similar previous studies and observed that their main focus was around advanced CKD (Groups 3a, 3b, 4 and 5) with almost no data on early CKD stages. The novelty of our study relies upon understanding the metabolic phenomena which ocur in the early CKD stages (Groups 1 and 2) and to provide insights into disease progression.

  1. Major revision:

(1) Abstract: This study is a pilot study. Thus, the conclusion should be made conservatively.

- Thank you for this useful recommendation. We mentioned in the text that this is a pilot study and therefore future research is required. 

(2) Introduction: The references cited in the fourth paragraph were older. There are more papers published in the PubMed in recent 5 years.

-Thank you for pointing this issue to us. We identified more recent references and replaced the older ones.

(3) Methods: This section of 2.4. Statistical analysis should be described in details. What are the goals and full names of PLSDA, VIP, MDA, log2FC, and HMDB? Which values were considered statistically significant?

- Thank you for this useful recommendation. We added in section 2.4 Statistical analysis a more detailed description where the full names for the used statistical tools can also be found. In the last line of the results, we also noted that the metabolites considered to be statistically significant were based on their statistical significance (p<0.05) correlated with data from the literature. The classification of these molecules was made using their retention time, m/z values, and peak intensities.   

(4) Figures 1-4: too small and unclear to read!

- Thank you for pointing this issue to us. We replaced them with higher resolution ones.

(5) References: less references in recent 5 years

- Thank you for your observation. We revised the references and added newer ones where possible.

Round 2

Reviewer 2 Report

The authors had well response to my questions. The manuscript can be accepted for publication in its present form with no revisions.